# A Novel Analysis of Super-Resolution for Born-Iterative-Type Algorithms in Microwave Medical Sensing and Imaging

**DOI:** 10.3390/s24010194

**Published:** 2023-12-28

**Authors:** Yahui Ding, Zheng Gong, Hui Zhang, Yifan Chen, Jun Hu, Yongpin Chen

**Affiliations:** 1School of Electronic Science and Engineering, University of Electronic Science and Technology of China, Chengdu 611731, China; dingyahui@std.uestc.edu.cn (Y.D.); hujun@uestc.edu.cn (J.H.); ypchen@uestc.edu.cn (Y.C.); 2Yangtze Delta Region Institute (Quzhou), University of Electronic Science and Technology of China, Quzhou 324003, China; zheng.gong@csj.uestc.edu.cn; 3School of Life Sciences and Technology, University of Electronic Science and Technology of China, Chengdu 611731, China; zhang.hui@std.uestc.edu.cn

**Keywords:** super-resolution, electromagnetic inverse scattering, Sparrow criterion, anti-apodization, Born iterative methods

## Abstract

Microwave medical sensing and imaging (MMSI) is a highly active research field. In MMSI, electromagnetic inverse scattering (EIS) is a commonly used technique that infers the internal characteristics of the diseased area by measuring the scattered field. It is worth noting that the image formed by EIS often exhibits the super-resolution phenomenon, which has attracted much research interest over the past decade. A classical perspective is that multiple scattering leads to super-resolution, but this is subject to debate. This paper aims to analyze the super-resolution behavior for Born-iterative-type algorithms for the following three aspects. Firstly, the resolution defined by the traditional Rayleigh criterion can only be applied to point scatterers. It does not suit general scatterers. By using the Sparrow criterion and the generalized spread function, the super-resolution condition can be derived for general scatterers even under the Born approximation (BA) condition. Secondly, an iterative algorithm results in larger coefficients in the high-frequency regime of the optical transfer function compared to non-iterative BA. Due to the anti-apodization effect, the spread function of the iterative method becomes steeper, which leads to a better resolution following the definition of the Sparrow criterion mentioned above. Thirdly, the solution from the previous iteration, as the prior knowledge for the next iteration, will cause changes in the total field, which provides additional information outside the Ewald sphere and thereby gives rise to super-resolution. Comprehensive numerical examples are used to verify these viewpoints.

## 1. Introduction

Microwave medical sensing and imaging (MMSI) has been an important research topic in the past decade. Due to the harmfulness of CT imaging to the human body, using microwaves for medical examination is a promising technology [1,2,3,4,5,6]. The applications of MMSI include the classification of stroke types, detection of early breast cancer, and others [7,8]. Electromagnetic inverse scattering (EIS) is a commonly used technique in MMSI, which aims at retrieving the interior distribution of relative permittivity in human organs.

For the performance of an imaging system, a key indicator is its resolution. The images obtained by EIS methods have exhibited the super-resolution characteristic, which has attracted the attention of many researchers. Generally, when the resolution of the image generated exceeds the diffraction limit (i.e., half of the carrier wavelength), it can be regarded as super-resolution. For instance, the resolution achieved in [9,10] is superior to 1/12 of the wavelength. This remarkable result goes beyond the diffraction limit, demonstrating exceptional super-resolution performance. Nevertheless, the rationale behind super-resolution remains an unresolved issue and the existing theoretical analyses often fail to explain the observed outcome. For example, Ref. [11] yields a resolution of 0.3 times the wavelength for the Born approximation (BA) algorithm, which contradicts the super-resolution phenomenon observed. Clarifying the underlying principle for super-resolution in EIS could help us to use the inverse algorithms better. On the one hand, we can know the scope of use of the inverse algorithm by analyzing the resolution. On the other hand, it could help us to develop more effective algorithms in the future.

A classical view is that super-resolution is due to the conversion from evanescent waves to propagating waves generated by the nonlinearity of EIS [12]. However, there are some controversies around this argument. According to [13], it is suggested that the presence of information outside the Ewald sphere in the far field does not naturally result in super-resolution. This is due to the fact that while multiple scattering can convert evanescent waves into transmission waves, the latter are ultimately destroyed, making it difficult to obtain the spectrum of the former due to the complex nature of scattering behaviors. Furthermore, a recent study by [14] shows that such multiple scattering would give rise to 0/0 manifolds from a mathematical perspective, making the achievement of super-resolution impossible. Another school of thought is that computational imaging is different from optical imaging; the former cannot directly use the resolution defined by the diffraction limit. Several new concepts have been introduced in this regard, such as the Cramér–Rao bound [15] and the average spatial resolution [16]. These methods apply the information-theoretic analysis to elaborate on the super-resolution behavior, but they mainly focus on the impact of noise, and thus, can only qualitatively explain super-resolution. Recently, several studies have indicated that the measurement configurations and parameters can have an impact on the resolution in EIS. For example, Ref. [17] focuses on the influence of background media on the singular values of the Green’s function; Ref. [18] focuses on the influence of the number of plane waves on the degrees of freedom; both affect the imaging resolution. Nevertheless, these analyses are not directly related to the underlying principles of super-resolution. In recent years, super-oscillation has also been used to explain the far-field super-resolution in EIS. The super-oscillation theory allows the spatial spectrum associated with the propagating waves to be used but still achieves resolutions beyond the diffraction limit. Several different methods are proposed to overcome the diffraction limit using super-oscillation [19,20].

In our previous work [21], we quantitatively analyzed the resolution of the BA by revisiting the definition of resolution. The key idea is to calculate the resolution using the Sparrow criterion and the generalized spread function. This yields the observation that larger cylinders are easier to separate. When the cylinder’s radius exceeds 0.048λ, the resolution comes to 0.25λ. Similarly, the new definition can be applied to iterative algorithms, which provides critical insight into the super-resolution analysis to be discussed in this paper.

Furthermore, we put forward another two interpretations of resolution improvement for iterative methods. Firstly, according to the simulation and experimental results, the resolution of the Born-iterative-type algorithms is much better than the BA. It is commonly believed that iterations decouple multiple scattering and result in a “full-wave” operation [22]. On the contrary, we propose a different interpretation of the role of iterations. The BA can be considered as a low-pass filter [11]. With the incorporation of an iterative process, this effectively results in a rearrangement of filter coefficients. To demonstrate this, we perform the Fourier transform of the point spread function (PSF). It can be seen that as the number of iterations increases, the PSF does not add any new information, but the high-frequency regime has larger coefficients. Due to the anti-apodization effect [23], the PSF becomes steeper. By applying the revisited resolution definition using the Sparrow criterion mentioned above, two scatterers can be separated more easily. This also explains why the Born-iterative-type algorithms are more effective than the BA. Secondly, the solution from the previous step is used as the prior knowledge for the next step, which changes the computed total field and brings additional information beyond the Ewald sphere. With sufficient prior knowledge, the PSF will change and the resolution can be improved. We will use a specific example to demonstrate this point.

The paper is organized as follows. In Section 2, we first present the inverse scattering model and the Born-iterative-type inverse algorithms, including the Born iteration method (BIM) and the distorted Born iterative method (DBIM). Next, because it is unable to calculate the PSF of BIM or DBIM as the total field keeps updating in the iteration process, we propose a “simplified” Born iteration method (SBIM) for PSF computation. Lastly, we explicate the super-resolution from the three aspects mentioned above, namely, the Sparrow criterion, larger coefficients of the high-frequency regime of PSF, and added information outside the Ewald sphere, using the BA, SBIM, and BIM/DBIM, respectively. Section 3 uses comprehensive numerical examples to verify our points. Finally, some concluding remarks are provided in Section 4.

## 2. Theoretical Analysis

### 2.1. Solutions of Born-Iterative-Type Algorithms

In this paper, we focus on a two-dimensional (2D) scenario that is illuminated by transverse magnetic waves. As demonstrated in Figure 1, the unknown scatterers are situated within the domain of interest (**D**). For simplicity, the antennas are assumed to be evenly distributed along the circumference of a circle (**S**). The background is homogeneous. The relationship between the scatterers and the electric field can be written as in the following equations [24]:(1)$$J=\chi (E^{i}+G^{D}J),$$
and
(2)$$E^{s}=G^{S}J,$$
where J is the contrast current; χ is the contrast function, defined as $\chi \left(r\right)=\left[\epsilon \left(r\right)-\epsilon_{0}\right]/\epsilon_{0}$, with ϵ(r) and $\epsilon_{0}$ being the permittivities of scatterer r and the background, respectively; $E^{i}$ is the incident field; $G^{D}$ and $G^{S}$ are the matrix forms of the Green functions in the domain **D** and the surface **S**, respectively; and $E^{s}$ is the scattering field. It is worth mentioning that our focus is on the resolution generated in the far field, thus the receiving antenna is located in the far-field area.

In general, the incident field $E^{i}$, the scattering field $E^{s}$, and the Green’s function matrices $G^{D}$ and $G^{S}$ are known. The contrast function χ and the current J are unknown. To solve the inverse scattering problems, a classical method is to use the BA to linearize the relationship between the scattered field and the contrast function. To apply BA, the scattered field must be significantly weaker than the incident field. Hence, the approximation $J=\chi E^{i}$ can be used. The two solving formulas are as follows:(3)$$J^{sol}={\left(G^{S}\right)}^{*}{\left[G^{S}{\left(G^{S}\right)}^{*}\right]}^{-1}E^{s},$$
and
(4)$$\chi^{sol}=\frac{\sum_{n=1}^{N^{i}}J_{n}^{sol}\cdot {\left(E_{n}^{i}\right)}^{*}}{\sum_{n=1}^{N^{i}}E_{n}^{i}\cdot {\left(E_{n}^{i}\right)}^{*}},$$
where ${\left(\cdot \right)}^{*}$ denotes the Hermitian conjugate, the subscript *n* denotes the nth incident wave, and $N^{i}$ represents the total number of incident waves. Following the derivation in [21], we can draw the conclusion that the solution of $\chi^{sol}$ is proportional to the convolution of χ and ${\left[J_{0}\left(k_{0}x\right)\right]}^{2}$,
(5)$$\chi^{sol}\left(r\right)=\frac{k_{0}\Delta R}{4\pi^{2}}\cdot \left\{\chi \left(r\right)*{\left[J_{0}\left(k_{0}r\right)\right]}^{2}\right\},$$
where $k_{0}$ is the wave number of the incident wave, ΔR is a constant accounting for the quantization effect in computation, and $J_{0}\left(\cdot \right)$ is the zeroth-order Bessel function of the first kind.

For the BIM method, the BA process is repeated multiple times using the result in the previous step as the prior information in the next step. Specifically, the solution of the current in (Equation 3) is changed to the residual of the current upon utilizing the residual of the scattered field,
(6)$$\Delta J^{sol}={\left(G^{S}\right)}^{*}{\left[G^{S}{\left(G^{S}\right)}^{*}\right]}^{-1}\left(E^{s}-E_{\chi_{n-1}}^{s}\right),$$
where $E_{\chi_{n-1}}^{s}$ is the scattered field in the previous iteration. Then, we can obtain the contrast function Δχ by $\Delta J^{sol}$. In consideration of each step, the change in the total field is very minor. Therefore, it can be approximated as $\Delta J=\Delta \chi E_{\chi_{n-1}}^{t}$, where $E_{\chi_{n-1}}^{t}$ is the total field in the previous iteration. This equation is highly overdetermined, rendering an exact solution impossible. A common approach is to obtain the least squares solution.
(7)$$\Delta \chi^{sol}=\frac{\sum_{m=1}^{N^{i}}\Delta J_{m}^{sol}\cdot {\left(E_{m}^{t}\right)}^{*}}{\sum_{m=1}^{N^{i}}E_{m}^{t}\cdot {\left(E_{m}^{t}\right)}^{*}},$$
where the subscript *m* denotes the mth incident wave, $N^{i}$ represents the total number of incident waves, and $E^{t}$ is the simplified representation of $E_{\chi_{n-1}}^{t}$. The main difference with the BA is that $E^{i}$ is replaced by $E^{t}$, which leads to additional information outside the Ewald sphere because $E_{\chi_{n-1}}^{t}$ incorporates the knowledge of $\chi_{n-1}$. Then, we can obtain the contrast function in the current iteration,
(8)$$\chi_{n}=\chi_{n-1}+\Delta \chi^{sol}.$$

For the DBIM method [25], the difference is that the Green’s function with background is used to obtain the current,
(9)$$G_{\chi_{n-1}}^{S}=G^{S}{\left(I-G^{D}\chi_{n-1}\right)}^{-1},$$
and
(10)$$\Delta J^{sol}={\left(G_{\chi_{n-1}}^{S}\right)}^{*}{\left[G_{\chi_{n-1}}^{S}{\left(G_{\chi_{n-1}}^{S}\right)}^{*}\right]}^{-1}\left(E^{s}-E_{\chi_{n}-1}^{s}\right).$$

### 2.2. PSFs of Born-Iterative-Type Algorithms

Assume that $\chi_{p}$ is the result from the last iteration. We can obtain the field function and the data function as
(11)$$J_{p}=\chi_{p}\left(E^{i}+G^{D}J_{p}\right),$$
and
(12)$$E_{p}^{s}=G^{S}J_{p}.$$
Then, we have
(13)$$J-J_{p}=\chi \left(E^{i}+G^{D}J\right)-\chi_{p}\left(E^{i}+G^{D}J_{p}\right),$$
and
(14)$$E^{s}-E_{p}^{s}=G^{S}\left(J-J_{p}\right).$$
We define $\Delta J=J-J_{p}$ and $\Delta \chi =\chi -\chi_{p}$, which yields
(15)$$\Delta J=\Delta \chi E^{i}+\chi_{p}G^{D}\Delta J+\Delta \chi G^{D}J_{p}+\Delta \chi G^{D}\Delta J,$$
and
(16)$$E^{s}-E_{p}^{s}=G^{S}\Delta J.$$
However, it is hard to solve the function with the most general form of ΔJ in (Equation 15). Various levels of simplification of the formula can be implemented as follows.

First, assuming that the changes in the total field are minimal in each iteration. So the ΔJ is far less than $J_{p}$. Hence, the second and fourth terms on the right-hand side of (Equation 15) can be ignored, thus it comes to
(17)ΔJ=ΔχEi+χpGDΔJ+ΔχGDJp+ΔχGDΔJ,
and the state and data functions can be written as
(18)$$\Delta J=\Delta \chi E^{i}+\Delta \chi G^{D}J_{p},$$
and
(19)$$E^{s}-E_{p}^{s}=G^{S}\Delta J.$$
And the solutions are expressed as
(20)$$\Delta J^{sol}={\left(G^{S}\right)}^{*}{\left[G^{S}{\left(G^{S}\right)}^{*}\right]}^{-1}\left(E^{s}-E_{p}^{s}\right),$$
and
(21)$$\Delta \chi^{sol}=\frac{\sum_{m=1}^{N^{i}}{\Delta J}_{m}^{sol}\cdot {\left(E_{m}^{t}\right)}^{*}}{\sum_{m=1}^{N^{i}}{E^{t}}_{m}\cdot {\left(E_{m}^{t}\right)}^{*}}.$$
We can find that these exactly correspond to the BIM in (Equation 6) and (Equation 7).

Next, if we use a better approximation and only ignore the fourth term in (Equation 15), it comes to
(22)ΔJ=ΔχEi+χpGDΔJ+ΔχGDJp+ΔχGDΔJ.
We define $\Delta J_{D}=\left(I-\chi_{p}G^{D}\right)\Delta J$, and the corresponding state and data functions are
(23)$$\Delta J_{D}=\Delta \chi E^{i}+\Delta \chi G^{D}J_{p},$$
and
(24)$$E^{s}-E_{p}^{s}=G^{S}{\left(I-\chi_{p}G^{D}\right)}^{-1}\Delta J_{D}=G_{\chi_{p}}^{S}\Delta J_{D}.$$
And the solution of $\Delta J^{sol}$ is given by
(25)$$\Delta J^{sol}={\left(G_{\chi_{p}}^{S}\right)}^{*}{\left[G_{\chi_{p}}^{S}{\left(G_{\chi_{p}}^{S}\right)}^{*}\right]}^{-1}\left(E^{s}-E_{p}^{s}\right),$$
which reduces to the DBIM. Hence, the reason why the DBIM is more effective than the BIM is because the DBIM uses a better approximation than the BIM.

However, it is difficult to find the closed-form solutions with either the BIM or the DBIM method, because $E^{t}$ in (Equation 2) is non-trackable. Subsequently, we propose the SBIM to derive the PSF by ignoring all the last three terms in (Equation 15).
(26)ΔJ=ΔχEi+χpGDΔJ+ΔχGDJp+ΔχGDΔJ,
The degenerated state and data functions are
(27)$$\Delta J=\Delta \chi E^{i},$$
and
(28)$$E^{s}-E_{p}^{s}=G^{S}\Delta J.$$
And the expression of $\Delta J^{sol}$ comes to
(29)$$\Delta J^{sol}={\left(G^{S}\right)}^{*}{\left[G^{S}{\left(G^{S}\right)}^{*}\right]}^{-1}\left(E^{s}-E_{\chi_{p}}^{s}\right),$$

In this case, it can be seen that every iteration step applies the BA, and the following PSF of the BA can be derived [21],
(30)$$\chi^{sol}=\frac{k_{0}\Delta R}{4\pi^{2}}\cdot \left\{\chi^{e}*{\left[J_{0}\left(k_{0}r\right)\right]}^{2}\right\},$$
where $\chi^{e}$ is the equivalent distribution of dielectric constants to radiate the residual of the scattered field. Subsequently, we can calculate $\chi^{e}$ as
(31)$$\Delta J=\chi^{e}\left(E^{i}+G^{D}\Delta J\right),$$
and
(32)$$\Delta E^{s}=G^{S}\Delta J.$$

The equivalent contrast function $\chi^{e}$ is
(33)$$\chi^{e}=\Delta \chi +\left(\chi_{p}G^{D}\Delta J+\Delta \chi G^{D}J_{p}\right){\left(E^{i}+G^{D}\Delta J\right)}^{-1}.$$

Finally, we can derive the progressive formula for the SBIM as
(34)$$\chi_{n}^{sol}=\sum_{i=1}^{n}\frac{k_{0}\Delta R}{4\pi^{2}}\cdot \left\{\chi_{i}^{e}*{\left[J_{0}\left(k_{0}\rho \right)\right]}^{2}\right\}.$$

Figure 2 shows the sectional views of the reconstructed images of point scatterers using different algorithms in 10 and 20 iterations, respectively. We can find that the results of the three algorithms (SBIM, BIM, DBMI) are the same because the nonlinearity is weak.

However, if the nonlinearity becomes stronger, the accuracy of the inverse algorithms will reduce. The resolution becomes irrelevant in this situation. For example, we set the frequency to be 300 MHz. A cylinder of radius 0.3 m is employed as the scatterer in this case. Figure 3 plots the sectional views of the reconstructed images with different relative permittivity values. In Figure 3a, the relative permittivity is set to 1.5; the reconstructed images are similar for all three methods. In Figure 3b, the relative permittivity is set to 2.5; the SBIM method fails in this case. Finally, in Figure 3c, the relative permittivity is set to 3.5; both the SBIM and BIM fail. The results demonstrate that we should avoid situations where the nonlinearity is too strong.

### 2.3. Explanation of Super-Resolution for BA and Born-Iterative-Type Algorithms

In this section, we will explain the super-resolution phenomenon from three different perspectives, namely, the Sparrow criterion, increased high-frequency components of PSF, and extra information outside the Ewald sphere by using the BA, SBIM, and BIM/DBMI algorithms, respectively.

#### 2.3.1. BA

The super-resolution for the BA lies in its definition of resolution based on the Sparrow criterion, which is defined as the separation distance between two targets when the joint function has no dip in intensity at the midpoint. Consequently, the resolution for point scatterers is
(35)$$\sigma =min\left|arg_{x}\left\{\frac{d^{2}\chi^{sol}\left(x\right)}{dx^{2}}=0\right\}\right|,$$
and the resolution for cylinder scatterers is
(36)$$\sigma =min\left|arg_{x}\left\{\frac{d^{2}\chi^{sol}\left(x\right)}{dx^{2}}=0\right\}\right|-2a,$$
where *a* is the distance from the edge to the center of each scatterer.

By using these two formulas, we can readily determine the criterion for super-resolution. In the case of BA, an analytical solution can be obtained [21]. Specifically, we found that the scatterer itself also affects the resolution of the imaging system. When the radius of the scatterer reaches 0.048λ, the super-resolution phenomenon can be observed. More detailed discussions can be found in [21].

#### 2.3.2. SBIM

It is widely recognized that the resolution of a Born-iterative-type algorithm is better than that of the BA. Because increasing the number of iterations cannot mitigate the nonlinearity in EIS [26], we hereby use the SBIM to explain the effect of multiple iterations on the shape of the PSF, which in turn influences the resolution.

First, we plot the sectional views of the PSFs of the SBIM in Figure 4a. It can be seen that the PSF of the SBIM becomes steeper as the number of iterations increases. Then, to prove that the SBIM does not provide information outside the Ewald sphere, we plot the sectional views of the 2D Fourier transforms of the PSFs (also called OTFs in Fourier optics) in Figure 4b. As shown in the figure, with the increase in the number of iterations, the OTF does not become wider, which indicates that the available information is still confined within the Ewald sphere. Nevertheless, the coefficients of the high-frequency components in OTFs have larger values. According to the PSF plots, the location of the first null remains the same even as the number of iterations increases. As such, the resolution based on the Rayleigh criterion also does not change. On the other hand, the PSFs become steeper as a result of larger high-frequency components in the corresponding OTFs, which is called anti-apodization in Fourier optics [23] and leads to improved resolutions following the Sparrow criterion. According to the definition of the Sparrow criterion, the numerical value of resolution is the distance of two points when the midpoint of the joint spread function does not decrease. It is equivalent to the second-order derivative of the joint spread function being zero. So the first zero point of the second-order derivative gives rise to the resolution limit. Subsequently, we provide the second-order derivative of the joint PSF of the SBIM in 1 iteration and 20 iterations in Figure 5. The first zero point is located at about 0.34 m in the SBIM in 1 iteration, while the point is at about 0.32 m in 20 iterations. Hence, the resolution is improved by the anti-apodization effect.

#### 2.3.3. BIM and DBIM

The reason why the BIM and the DBIM perform better than the SBIM is that they use the total field to replace the incident field, which utilizes extra information outside the Ewald sphere.

Consider the BIM as an example. The first iteration in the SBIM is identical to that in the BIM. However, in the second iteration, although the SBIM and the BIM have the same initial values, they yield different results. The only difference between the SBIM and the BIM is the field in the reconstruction formula based on (4) and (7). First, we take the Fourier transform of the least squares formula of the SBIM. Generally, the incident field is a plane wave or can be converted to a plane wave from other types of waves. The Fourier transform of a plane wave is a point located at $\left(k_{0},\theta \right)$ (polar coordinate) in the spectral domain, as illustrated in Figure 6a. Points in different directions will form a ring. The Fourier transform of the ring is the zeroth-order Bessel function. So the least squares formula by the plane wave is a convolution of the zeroth-order Bessel function. When the incident field is replaced by the total field in the least squares formula of the BIM, the points in the spectral domain would spread to a complex region surrounding the original point, as depicted in Figure 6b, which is dependent on the total field related to the prior information. As such, some information outside the Ewald sphere would be incorporated in the inverse process, effectively extending the original OTF and enhancing the resolution.

## 3. Numerical Results

In this section, we will use comprehensive numerical examples to validate our points.

### 3.1. Effect of the Sparrow Criterion on Resolution in BA

First, we investigate the resolutions of point and cylinder scatterers in the BA algorithm. Figure 7 and Figure 8 are the reconstructed images of point scatterers and cylinder scatterers by using the BA. And those figures also can be found in [21].

In Figure 7, the actual images are given in Figure 7a,d,g. The distances of the two points are 0.30 m, 0.34 m, and 0.38 m, respectively. Then, we give the reconstructed images in Figure 7b,e,h. To confirm whether the two points are distinguished, we give the sectional views of the reconstructed images in Figure 7c,f,i, according to the sectional views. There is a slight trough between the two peaks in Figure 7i. So the two points are resolved at a distance of 0.38 m. But it cannot be resolved at a distance of 0.34 m. In addition, we use the signal-to-peak-interference ratio (SPIR) to estimate the degree of the difference. The SPIR is defined as the ratio of the intensity of the recovered signal (reconstructed images at the actual scatters) and the intensity of the peak interference (reconstructed images at the midpoint):(37)$$SPIR=\frac{\epsilon_{act}-\epsilon_{0}}{\epsilon_{mid}-\epsilon_{0}},$$
where $\epsilon_{mid}$ and $\epsilon_{act}$ are the reconstructed relative permittivity at the midpoint and the actual scatters, respectively. $\epsilon_{0}$ is the relative permittivity of the background. Apparently, a bigger SPIR means a better resolution. The two scatters can be distinguished when the SPIR is more than 1. The SPIRs of Figure 7c,f,i are 0.8308, 0.8475, and 0.8775, respectively. The SPIR of Figure 7i is less than 1 because the reconstructed relative permittivity of the highest point is not the actual scatter position. It should be regarded as a special case.

For cylinder scatterers, the frequency and cylinder radius are set to 300 MHz and 0.1 m, respectively. The resolution is calculated as 0.152 m using Equation (Equation 36). Figure 8a,b show the actual and reconstructed images for a separation distance of 0.12 m. Figure 8d,e are the images for a separation distance of 0.16 m. Figure 8g,h are the images for a separation distance of 0.2 m. Figure 8c,f,i are the sectional views of the reconstructed images. It is easy to determine that the two cylinders are distinguished when the distance is 0.2 m. However, it is difficult to determine whether the two cylinders are resolved or not at a separation distance of 0.16 m due to their closeness to the computed resolution limit. But we believe that this distance is the dividing line that can be distinguished. According to the calculation, the resolution limit with respect to wavelength is about 0.152λ. Therefore, the super-resolution in EIS is explained by the Sparrow criterion. The SPIRs of Figure 8c,f,i are 0.8523, 0.9488, and 1.1045, respectively.

### 3.2. Effect of Larger High-Frequency Components of PSF on Resolution in SBIM

Next, we look into the same scattering scenarios using the SBIM with 20 iterations.

In Figure 9, we present the reconstructed images and the sectional views of two points with separation distances of 0.30 m (Figure 9a–c), 0.32 m (Figure 9d–f), and 0.34 m (Figure 9g–i). The critical distance is reduced from 0.34 m to 0.32 m. We can find that the resolution is improved in the SBIM compared to the BA due to the increased coefficients of the high-frequency components of the PSF. The SPIRs of Figure 9c,f,i are 0.8908, 0.9602, and 1.0666, respectively.

### 3.3. Effect of the Prior Knowledge on Resolution in BIM

Finally, we design a specific scenario to demonstrate the effect of the prior knowledge on the resolution in the BIM.

Figure 10a shows two points located in a cuboid. The distance of the points is 0.32 m, and the frequency is set as 300 MHz. As known, the two points cannot be separated using the SBIM method. On the other hand, we add a cuboid in this situation, which is supposed to be known before applying the inverse algorithm. Figure 10b depicts the cuboid. Figure 10c plots the Fourier transform of the incident field when θ=0. Figure 10d shows the reconstruction image of SBIM. Figure 10f is the Fourier transform of the total field by the cuboid. Figure 10g is the reconstruction image of the BIM. Figure 10e,h are the sectional views of Figure 10d,g, respectively. According to Figure 10f, the Ewald sphere is expanded, and the resolution is improved accordingly, as shown in Figure 10h. The SPIRs of Figure 10e,h are 0.9582 and 1.0487, respectively. Compared to Figure 9, we can find that adding a cuboid with known properties has a positive effect compared to not having that cuboid but knowing the properties of the background medium. This is also because the total field is changed so that the information outside the Ewald sphere is calculated in the inverse process. The OTF is enlarged and the resolution is enhanced.

## 4. Conclusions

In this paper, we have analyzed in detail the sources of super-resolution for Born-iterative-type algorithms. First, using the Sparrow criterion instead of the traditional Rayleigh criterion could clarify the impact of different factors in the super-resolution of the inverse algorithm. The Sparrow criterion is defined as the separation distance between two targets when the joint function has no dip in intensity at the midpoint. Then, we propose the SBIM to divide the contribution of two factors on the super-resolution of the iterative method. The SBIM is a simplified method that ignores more terms compared to the BIM and the DBIM. Then, the rationales behind the super-resolution have been summarized into three aspects. The first aspect is that the size of the scatter also has an impact on the resolution.

The last two aspects are the key contributions of this paper. First, iterative algorithms have larger coefficients in the high-frequency part of the OTF compared to the BA. Due to the anti-apodization effect, the PSF of an iterative method becomes steeper, which leads to a better resolution under the Sparrow criterion. Second, as the iteration progresses, the solution from the previous iteration as the prior knowledge will bring about changes in the total field, which gives rise to added information outside the Ewald sphere for resolution improvement.

The limitation of this paper is that we have not analyzed the impact of nonlinearity on the super-resolution. The shape and position of the PSF would be distorted and the definition of super-resolution would become invalid if the nonlinearity was very strong. The effect of nonlinearity on the resolution would be an interesting study in the future.

## Figures and Tables

**Figure 1 sensors-24-00194-f001:**
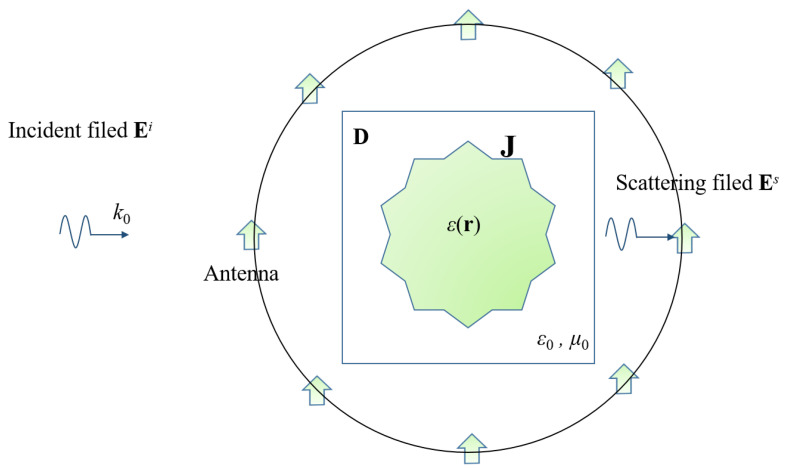
Schematic diagram of EIS.

**Figure 2 sensors-24-00194-f002:**
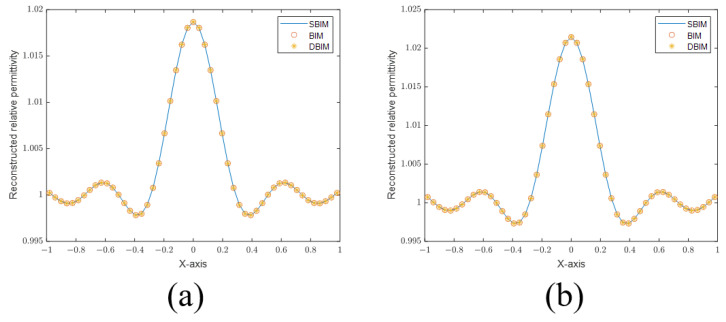
(**a**) Sectional views of reconstructed images of a point scatterer in 10 iterations. (**b**) Sectional views of reconstructed images of a point scatterer in 20 iterations.

**Figure 3 sensors-24-00194-f003:**
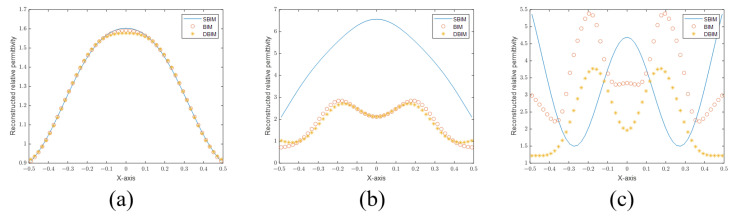
(**a**) Sectional views of reconstructed images of a cylinder with a dielectric constant of 1.5. (**b**) Sectional view of the reconstructed image of a cylinder with a dielectric constant of 2.5. (**c**) Sectional views of the reconstructed image of a cylinder with a dielectric constant of 3.5.

**Figure 4 sensors-24-00194-f004:**
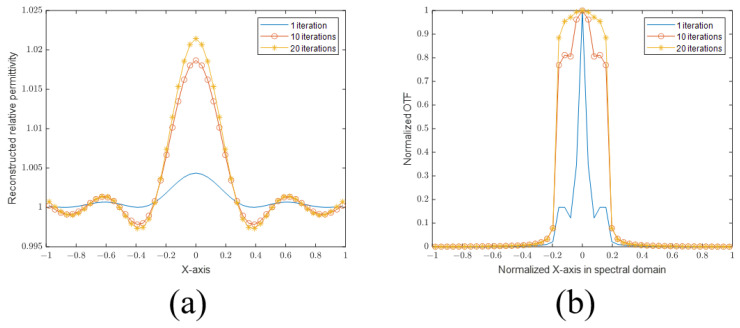
(**a**) PSFs with 1, 10, and 20 iterations. (**b**) Normalized OTFs with 1, 10, and 20 iterations.

**Figure 5 sensors-24-00194-f005:**
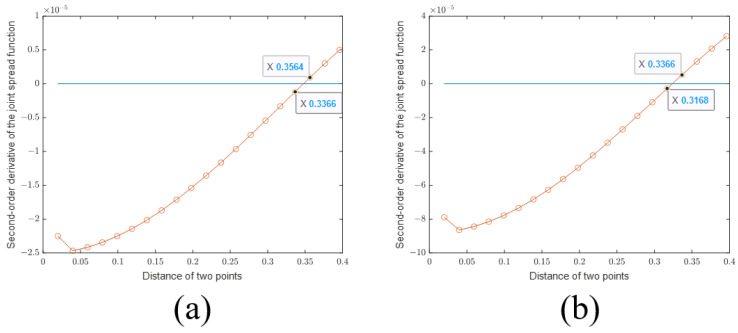
Second-order derivative of the joint PSF of SBIM with (**a**) 1 and (**b**) 20 iterations.

**Figure 6 sensors-24-00194-f006:**
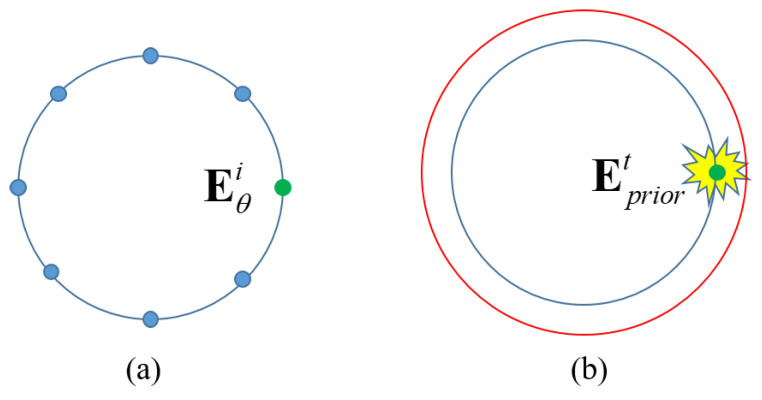
Pictorial illustrations of OTFs in (**a**) SBIM and (**b**) BIM.

**Figure 7 sensors-24-00194-f007:**
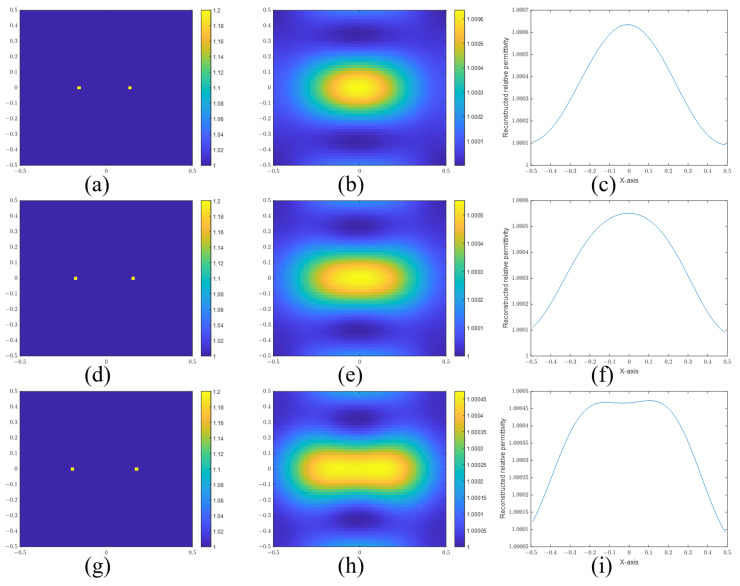
(**a**) Real image, (**b**) reconstructed image, and (**c**) sectional view of two points with a separation distance of 0.30 m. (**d**) Real image, (**e**) reconstructed image, and (**f**) sectional view of two points with a separation distance of 0.34 m. (**g**) Real image, (**h**) reconstructed image, and (**i**) sectional view of two points with a separation distance of 0.38 m.

**Figure 8 sensors-24-00194-f008:**
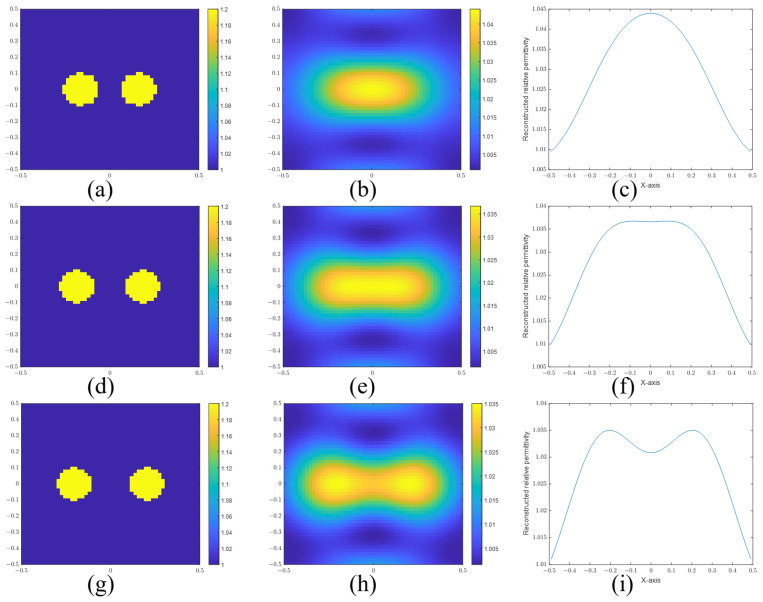
(**a**) Real image, (**b**) reconstructed image, and (**c**) sectional view of two cylinders with a separation distance of 0.12 m. (**d**) Real image, (**e**) reconstructed image, and (**f**) sectional view of two cylinders with a separation distance of 0.16 m. (**g**) Real image, (**h**) reconstructed image, and (**i**) sectional view of two cylinders with a separation distance of 0.20 m.

**Figure 9 sensors-24-00194-f009:**
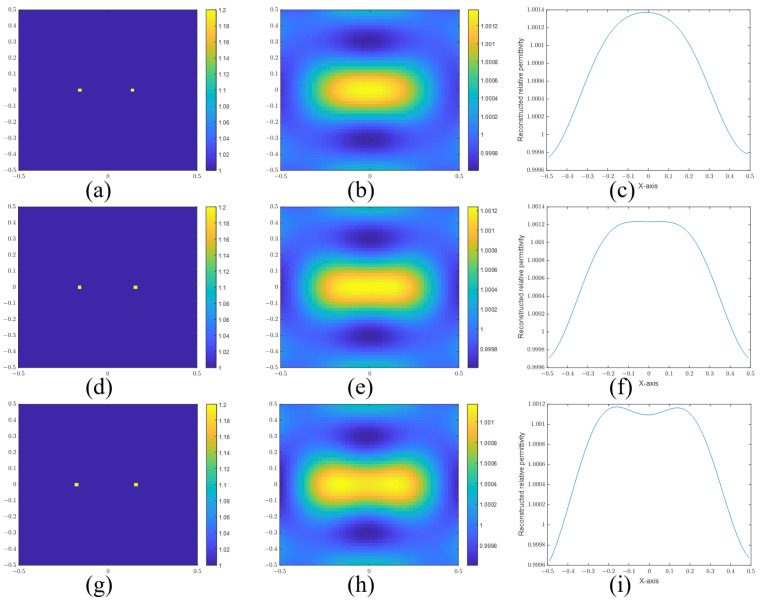
(**a**) Real image, (**b**) reconstructed image, and (**c**) sectional view of two points with a separation distance of 0.30 m. (**d**) Real image, (**e**) reconstructed image, and (**f**) sectional view of two points with a separation distance of 0.32 m. (**g**) Real image, (**h**) reconstructed image, and (**i**) sectional view of two points with a separation distance of 0.34 m. SBIM with 20 iterations is employed in this case.

**Figure 10 sensors-24-00194-f010:**
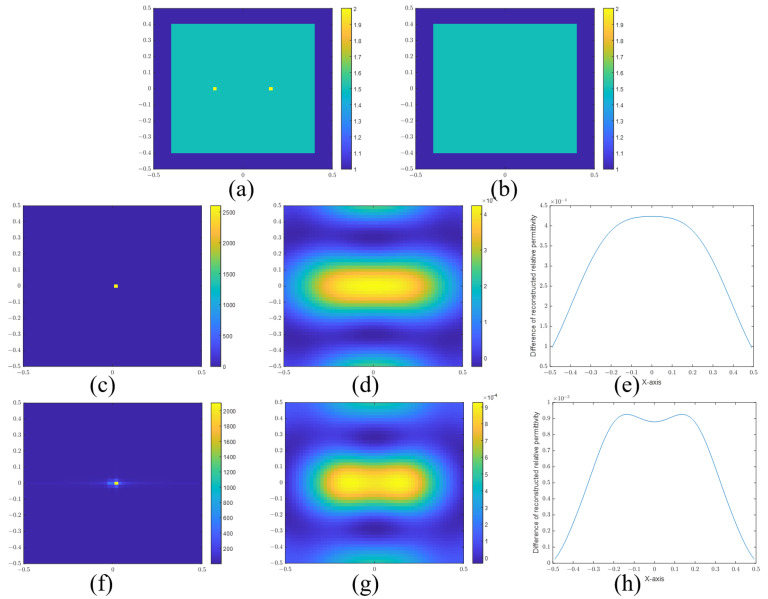
(**a**) Actual permittivity profile of two point scatterers and the (**b**) square cylinder. (**c**) Fourier transform of the incident field when θ=0. (**d**) Reconstruction image of SBIM and (**e**) sectional view. (**f**) Fourier transform of the total field by the cuboid. (**g**) Reconstruction image of BIM and (**h**) sectional view. The BIM algorithm is employed in this case.

## Data Availability

Data are contained within the article.

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
