# Peer review of "A Novel Analysis of Super-Resolution for Born-Iterative-Type Algorithms in Microwave Medical Sensing and Imaging"

_sensors, 2023, doi:10.3390/s24010194_

Round 1
Reviewer 1 Report
Comments and Suggestions for Authors
This manuscript proposes a novel analysis of super-resolution for born-iterative-type algorithms in microwave medical sensing and imaging. As a whole, this manuscript shows great potential to capture the attention of readers and specialists working in the field. To further improve the quality of the manuscript, some suggestions and corrections should be addressed.
1.The motivation of this manuscript should be further enhanced so that readers can better understand it.
2.Please list key contribution of this work and include discussion of limitation in conclusion section. Discussion of limitation is important, so reader understand boundary of this work.
3.Limited number of performance metrics are used. Additional relevant metrics may be used.
4.The introduction of related works is limited. It is strongly suggested that the authors considered more recently published research works, such as microwave medical sensing and imaging: dynamic low-rank and sparse priors constrained deep autoencoders for hyperspectral anomaly detection, and microwave medical sensing and imaging: deep self-representation learning framework for hyperspectral anomaly detection.
Comments on the Quality of English Language
Minor editing of English language required.
Reviewer 2 Report
Comments and Suggestions for Authors
The idea that the super-resolution can be related to the size of the scatterers looks promising. This paper can contribute to better explain which are the possible implications of the Born-iteration-algorithm in the implementation of super-resolution. This paper proposes a new perspective for interpreting the super-resolution phenomenon.
Reviewer 3 Report
Comments and Suggestions for Authors
Several parts of the paper need further explanation or supported by references. Please refer to the attached file.

Round 2
Reviewer 1 Report
Comments and Suggestions for Authors
This version is good for publication.
Reviewer 3 Report
Comments and Suggestions for Authors
Thank you for the revisions. Please modify this part in the revised manuscript:
"The super-oscillation theory enlarges the bandwidth of the observed data without involving evanescent waves."
Modify as:
"The super-oscillation theory allows to use the spatial spectrum associated with the propagating waves but still achieve resolutions beyond the diffraction limit."
Comments on the Quality of English Language
Minor editing would help to imprve the quality of manuscript.
